# Performativity of the Memory of Religious Places through Sound and Image

Frederico Dinis [1,2]

1 CEIS20—Centre for Interdisciplinary Studies, University of Coimbra, 3004-531 Coimbra, Portugal; f.dinis@sapo.pt
2 CITER—Research Centre for Theology and Religious Studies, Portuguese Catholic University, 1649-023 Lisboa, Portugal

**Abstract:** In this essay, we explore and deepen the confluence between sound and image, linking and relating concepts, purposes and coherence of artistic practices mediating and reconfiguring the memory of religious places. We observed that the performativity of memory, as an autobiographical concept, can be enhanced through live audiovisual performances in religious places. We have established that the performativity of memory in religious places can promote a spatial 'self', creating dynamic, immersive and physical experiences in the religious places. And we argue that the construction of this spatial 'self' involves processes of social and artistic reconfiguration that contribute to transforming not only the social dynamics within the community but also the artistic representations of memory. These main findings were reached following a process of research through artistic practice, thus a systematization of the processes involved in approaching three religious places. It also assumes the (de)construction of the sense of place throughout a personal reading on the mediation through nonverbal means. In this research, we also observed that the aesthetic and performative configurations can have an impact on the most individual manifestations of religion, religiosity and religious belief, influencing the interpretation and creation of meaning, evoking emotional and spiritual responses.

**Keywords:** sound; image; memory; religious places; site-specific; sense of place

## 1. Introduction

The concept of performance can be understood through different perceptions resulting from different disciplinary approaches, artistic fields or cultural contexts (Wardrip-Fruin 2006). It is also because of this conceptual openness and the diversity of creative practices that performance, as an action in front of an audience (Carlson 2004), offers a great potential for exploration, on which we would like to base this essay, focusing especially on the confluence between sound and image, mnemonic and sensory media, as a theme of theoretical and creative recognition, in religious places.

Performance has become a meeting point between the arts, where unconventional forms of dialogue converge in the desire to develop an experiment of fusion of artists with an intertextual, multisensory, technological and experiential approach to the event (Dinis 2020). In this sense, audiovisual performances are characterized by the inclusion of alternative aesthetics and technological innovation, decentralizing from the body/performer and extending to other media, namely sound and image.

The interrelationship between sound and visual media thus appears as a device to weave new possibilities of constructing meaning, where dialogical connections can emerge through rereading and reinterpretations that induce the audience to move to other contexts through the manipulation of sound and image, where the sound follows the image or vice versa, not homogeneously mixed in fruition. Sound and image, in confluence, offer a wider range of possibilities for interrelations, elaborating other narratives that, in the

performative moment, are no longer either sound or visual, but both, in a materialization of an ephemerality achieved through the use of technological means. We also assume that in the performative moments, the artist/performer is considered not only as an operator of sound and visual media but also as a mediator, a creator and, consequently, a narrator who constructs the sound and visual narratives.

Sound and image evolve in a combinatorial potential that conceptually expands the live audiovisual performance. And when sound and image simultaneously capture both senses in a single aesthetic-narrative sense, there is an articulation that not only captures the viewer's attention but also leads to new interpretations. This discussion of the capacity of live audiovisual performance to promote new interpretations and poetic readings of works raises new questions about the production of effects of presence through technological mediation in specific places. It is, therefore, necessary to examine these territories and the processes of mediation in the production of effects of presence in audiovisual performative moments in religious places.

Although other approaches reconstruct the spatial self through ritual and belief, this essay focuses on exploring the confluence of sound and image in religious places through a process of research through artistic practice in the field of contemporary performance, analyzing the role of site-specific audiovisual performances in the process of perception and apprehension, reflecting on the effects of the embodied perception of space through a performativity of memory in religious places, and examining the effects of the aesthetic and performative configurations through which religion/religiosity/religious belief is transmitted on its most individual manifestations in site-specific nonverbal artistic creations.

To respond to these objectives, a methodology of research through artistic practice is used, based on the principle that the performativity of memory and the construction of a narrative as a temporal text (Fonseca 1992) reinforces the role of sound and visual media in the context of audiovisual performances and during performative moments. The involvement of the audience thus escapes the commonplace of everyday corporeality, diluting the permanent boundaries and exploiting physical experience as a motto for spatial transgression and the construction of a spatial 'self'.

To rethink these issues and structure the arguments, this essay is divided into four main parts, in addition to this Introduction and the Conclusions at the end.

In the Section 2, Presence Effects, we consider the significance of the question of presence and its emptiness through technological mediation, creating in the debate an apparent opposition between the 'live' and the 'mediated'. We also argue that presence does not belong to a particular medium or living body but is produced through performative, live and mediated moments.

In the Section 3, Performative Moments, we reflect on the concepts of performative space and time in performative practice to analyze the performative moment as a moment of artistic expression that promotes a symbolic liturgy. We also highlight the conditions of space that affect performance, namely, in religious places as spaces that have special and intangible qualities, arguing that religious places are relational and contingent.

In the Section 4, Performativity of Memory, we analyze the place as a space endowed with sensations, affections and allusions to lived experiences and memories as lived records that start from memories and eternalize places as references and passages. We also highlight the importance of the performativity of memory that operates through sounds and images and which operates as an activator.

The Section 5, Methodology, presents the process of research through artistic practice (practice-as-research) followed in this research, guided by a 'conceptual model of approach to place' (Dinis 2020). It is thus a systematization of the processes involved in approaching the three religious places where the site-specific projects were developed, which is understood as a process of constant questioning. We also propose the (de)construction of the sense of place throughout a personal reading on the mediation through nonverbal means, focused on exploring key issues in this research.

Starting from a set of artistic practices and audiovisual performative moments that we have developed in this research, we explore and deepen the confluence between sound and image, linking and relating concepts, purposes and coherence of artistic practices that use these two means in the reformulation of religious places and how individual and community identities are mediated and reconfigured. Religious places are spaces that seek to create a place of spiritual connection and to reinforce religious ontological positions in the world, where interaction with the sacred is found and where the meaning and significance of human existence are intensified (Barrie 2010).

This intensification can be enhanced by mediation through sound and visual means, since when sound and image simultaneously capture both senses in a single aesthetic-narrative sense, there is an articulation that not only captures the viewer's attention but also leads to new interpretations (Wardrip-Fruin 2006).

These interpretations raise several issues about the perception of site-specific performances through the mediation of nonverbal means in religious spaces. How can the performativity of memory be an autobiographical concept enhanced through live audiovisual performances in religious places? How can the performativity of memory in religious places promote a spatial 'self' through live audiovisual performances? How does the construction of a spatial 'self' involve processes of social and artistic reconfiguration in live audiovisual performances in religious places compelled through the performativity of memory?

For the development of these lines of inquiry, practice needed to be present throughout the process, as the issues considered were the result of the practical component; otherwise, we would not have truly 'practice-based' research (Silva 2011). The approach followed in this research involved the realization of three site-specific projects in religious places (Tree of Life Chapel in Braga, Chapel of the Immaculate in Braga and Church of Cedofeita in Porto), under a proper format and language of research through artistic practice, intending to analyze and develop functional methods and strategies linked to the development of the artistic creations themselves and the proposal of ways of locally representing them.

These site-specific projects are also anchored in previously developed research through artistic practice and amplify nonverbal aesthetic and performative forms through which religion manifests and can be culturally transmitted. This previous research through artistic practice has shown that the interrelationship between sound and visual mediums, the performativity and the work of memory in site-specific performances emerges as an artifice to weave new transformations that inspire different forms of knowledge, whether intellectual, performative or sensory.

A website[1] has also been structured to present documentation on the process of designing, presenting and receiving site-specific projects. Taken as a whole, these materials deepen and illustrate the paths of research through artistic practice and are therefore materials inherent to creative making itself, rightly understood as a reflective practice.

## 2. Presence Effects

Over time, the evolution of technology and art has facilitated the blending of different art forms, allowing audiovisual artists to cultivate a distinct aesthetic centered on sound, image and 'in motion'. As a result, the synergy between sound and visual elements continues to be explored across disciplines, effectively broadening the conceptual scope of live audiovisual performance. This convergence of auditory and visual components enhances the immersive quality of 'live' experiences, creating a harmonious relationship that improves the overall aesthetic encounter.

The growing integration of new media and technologies into the performing arts has led to the emergence of diverse artistic expressions and methods. This phenomenon has also been documented in the theoretical field, and its influence can be measured by the large number of anthologies and texts that have been published on the subject in recent decades[2]. These approaches generally vary between understanding the use of new technologies and media as a rupture and as something new to be integrated into artistic practices or as a

continuity in the technological integration of artistic practices (Dinis 2021). For those who see this integration as a rupture, the use of technologies allows for the emergence and development of 'new artistic possibilities' (Saltz 2013, p. 422). Others, however, see these practices as the unfolding of a pre-existing potential, since "theatre has always used the most advanced technology of its time to enhance the 'spectacle' of productions" (Dixon 2007, p. 39).

These different visions reflect the construction of a critical discourse on the implications of the use of new media in transforming artistic forms and practices. One of the main issues raised in these discourses concerns the question of presence and its emptying through technological mediation, creating in the debate an apparent opposition between the 'live' and the 'mediated'.

The course of artistic movements associated with performance art, in the context of its history and its dominant languages, seeks to value the presence of the performer as something that can be experienced immediately, in the encounter between spectator and performer, and above all as the goal of the performance itself—that is, an absolute state of presence that Fischer-Lichte (2012) defines as 'radical presence', which means appearing and being perceived as an embodied spirit.

The discourse on performance tends to emphasize the character of 'live' art as its most distinctive feature, thus evading reproduction, as opposed to 'mediated' art, as Phelan (1993) argues. The ontology of performance is paradoxically enunciated in both presence and absence and in "all the conceptual oppositions of metaphysics" (Derrida 1988, p. 26). Auslander (1999) criticizes Phelan by arguing that the category of 'live' is actually an effect of mediation, not the other way around (Auslander 1999). Historically, 'live' only exists from the moment that reproduction techniques are invented, thus defining itself as what can be recorded. In this sense, the 'live' is linked to and dependent on mediation. Furthermore, Auslander (1999) notes that there is a progressive tendency to mediate 'live' events and understands the insistence of theoretical discourse on performance to situate it as a purely ideological, as opposed to a mediated or technological, form of art.

In this essay, we recognize that contrary to the emphasis on performance as something that takes place 'live', performance appears to be linked to various means and processes of mediation, mainly through the use of the image. From the perspective of those who value the 'live' character of the performance, mediation is seen as a factor that weakens the presence of the performer, but this impoverishment is not always seen as negative[3].

The growing mediation in the context of theater and performance marked the post-modern deconstruction of presence in the theoretical field through the decentralization of the subject and the narrative fragmentation that emptied[4] the authority of the text, creating a kind of persona that functions as a strategy of deconstruction of presence and structures of authority in performance (Auslander 1994). In addition to the emptying of the character, the deconstruction of presence is also due to the conceptual approach of the collective's work, which does not rely on the direct representation of a dramaturgical text but instead creates its texts and guides of performance[5].

The understanding of the use of new media as a strategy for deconstructing presence, defended by authors such as Pontbriand (1982), Fuchs (1985), Auslander (1999) or Féral (1985, 1992), has been reinforced by the valorization of the experience of presence in theater, performance and the arts in general with particular emphasis on the processes of producing presence through technological means. Thus, as Pontbriand (1982, pp. 155–56) argues, "presence no longer depends on materiality, but on the exhibition value of the work of art, its multiplicity and its accessibility". Presence, understood as 'being in front of', 'in the face of' something or someone who is 'other', always implies plurality and otherness and occurs in the dynamic between production and performance and in the reception of that presence.

Cusack (2007) also questions the increasingly dominant function of technology in performance, stating that the question is not what happens to the body and the living presence of the actor/performer but rather how to re-imagine the 'live' in a radically

networked digital world, where the experiences of presence for both the performer and the spectator are increasingly mediated.

Presence does not belong to a particular medium or living body but is produced through performative, live and mediated moments (Dinis 2020). The technological means of artistic expression allow the integration of different arts in the same work, and in this sense, the means allow dialogue and communication between different artistic realities. At the same time, these technological, audiovisual and/or multimedia media are auxiliary means of reproduction of the artistic object, becoming themselves artistic objects, distinct and autonomous forms of art, linked only by ties of descent from the other arts. Technological means, especially audiovisual media, must then be seen as a bridge between art objects and not as an artistic goal.

Audiovisual creations are an extension of the emotions and senses of the individual through media, making it possible to experience sensations that would otherwise be out of reach. The presence of performers is thus articulated in the relationship between live and mediated performance, in the tension between the isolation of bodies and technological encounter, in how physical and electronic presences are interdependent and complementary rather than exclusive or antagonistic and in the dynamics between performance and reception of that presence in the performative moments.

## 3. Performative Moments

In these performative moments, where the relationship of presence in a mediated performance is established, it is important to reflect on the concepts of performative space and time in performative practice that promotes a symbolic liturgy. A work of art can be completely liberated from space (Goldberg 2007), since all spaces in which any action can be performed attended by at least one person can be considered performative spaces (Rancière 2010; Alvarez 2004). However, in the case of performance, it is necessary to have an adjusted understanding of the conception of this performative space so that it can be assumed as such.

The experience of space is based on two conceptions linked with performative space: (i) the space that is conceived as a space that must be filled; (ii) the space that is conceived as invisible, unlimited and linked to its beneficiaries by coordinates, displacements and trajectories and observed as a substance to be extended. These two conceptions of space correspond to two different ways of describing it: the external objective space and the gestural space (Pavis 2003).

In addition to these conceptions of performative space, we can also highlight the multiplicity of other performative spaces and/or the adaptation of spaces with other functions, such as performance venues. These are conceptions of space that are intrinsically linked to a language of their own, where concepts such as performance and improvisation extend the limits of performance (Artaud 1996; Brook 2008; Rancière 2010).

This opens up new possibilities, recursive processes, repetitions, nonlinear structures, simultaneous events and a mixture of languages, where time, performative space and the performativity developed between the performer and the spectator are related with greater choice, with the cooperation of different means and often with the appropriation and invasion of new spatiality and in different performative moments (Dinis 2020).

These performative moments are the artistic expression in themselves, since they are the encounter of the performers, through their work, with the spectators and with their fruition. Thus, live audiovisual performance proposes the moment as an artistic expression that will always be a po(i)etic event (Duque 2018). Despite the possibilities of technologies for infinite repetition, the conceived performative moment is unique. This is because the quality of unrepeatability is not the result of chance or the unexpected, as is common between rehearsed and improvised performances. This quality is the result of the unrepeatable dynamic between the audience and the artists, in the way the latter react (albeit emotionally) to the sound and visual work of the performers, and also the dynamic between the artists themselves (Dinis 2020).

The conditions of the space also have an impact on the performance (Howell 2022), thanks to some specific qualities. The spaces that host these practices of the moment are of variable typology: gallery, theater, cinema hall and museum, but can also include outdoor spaces, and found spaces, among other spaces that are momentarily contextualized as artistic, such as religious places.

Religious places are spaces where special, intangible qualities can be revealed as a kind of quality that makes a place special (Barrie 2010). Qualities that are intertwined with the sense of place, the genius loci, which concerns fewer observable qualities in an environment, such as character and atmosphere (Norberg-Schulz 1988). These are spaces that seek to create a place of spiritual connection and reinforce the ontological religious positions in the world, where an interaction with the sacred is found and where the meaning and significance of human existence are intensified (Barrie 2010). These are also spaces that invite contemplation of the divine mystery in the built form and encourage a deeper understanding of the construction of place, our presence in the space and our role in human life. Space is not only transformed into place through meaning, and the religious place has an intangible meaning that is revealed in its spatial constructions (Norberg-Schulz 1979).

In this sense, religious places are relational and contingent, experienced and understood differently by different people, because they are multiple, contested, fluid and uncertain, producing multiple effects on the individual and provoking different transformations throughout live audiovisual performative moments.

During these live audiovisual performative moments in religious places, sound and image become convergent expressive processes, configuring a production of effects of presence, stimulating a representation of memory and promoting the creation of new immaterial meanings. The articulation between sound and image also promotes the creation of new narratives, making them denser and more immersive in artistic figuration, reflecting the complementarity of place and time, and opening new performative paths that are developed and experienced in live audiovisual performances.

## 4. Performativity of Memory

The notion of performativity was introduced into linguistic theory by the British philosopher John Langshaw Austin (1962) in his lecture series, "How to Do Things with Words", at Harvard University in 1955 and subsequently discussed by several authors[6].

The notion of performativity has also been developed within the field of performance studies. In this sense, the concept of performativity is elaborated at the moment when the performative act takes place, where performance and life intersect, in a sequential construction of different intersections where the effects of the real and the fictional are dissolved (Fernandes 2011). It is a space and time where the particularities of fiction, developed through performance, and the understandings of social life, guided by the real, come together in a 'doing' where their imaginary boundaries are transgressed.

It can also be said that the notions of performativity consider 'the other', the spectator, as a collaborator in the performative game. The spectator, in turn, can observe and be observed, affect and be affected, configuring an aesthetic experience characterized by open and procedural actions, thus promoting a resignification of experience (von Hantelmann 2014).

Performativity also develops the proposition of the actor/performer's own body as discourse (Austin 1962). For Féral (2015), artists who engage in and carry out performative actions are first and foremost generators of energy flows that transcend the notion of representation without fixing or focusing on it. As a border art, performance highlights the performer's body in all its fragility, autonomy and often in its insubordination to a script previously conceived by the performer himself, since the experience given by the performative moment can contaminate the content of the proposal, redirecting the performer to other possible places. Consequently, the interferences (or contaminations) of space, light, sound and audience mediate the performer's experience in space, unfolding the process of performing the work live, valuing the process over the notion of a finished product. This leads to another point emphasized by Féral, the 'total involvement of the

artist' (Féral 2015), in which the performer invests in a strong presence, not worrying about the formalisms of a message, but transforming his body into a discourse.

Performativity, in this sense, escapes the intention of traditional aesthetic theory, as it resists the hermeneutic disputes of understanding the work of art, falling within what Krauss (1990) has called a 'lived bodily perspective' or what Taylor (2022, p. 90) describes as "a process of becoming, of coming into being".

For Fischer-Lichte (2007), understanding the artist's actions is less important than experiencing them, crossing the proposed event. Participation in the experience provokes such a range of sensations that it transcends the possibility and effort of interpretation and the production of meaning and cannot be overcome or resolved by reflection. This is not to say that in a performance there is nothing for the spectator to interpret, but neither can it be said that the actions of the performance artist alone mean something.

The notion of performativity is linked to art as a network of exchange between artistic action and audience, guided not only by the sense of scenic representation but also by the approximation of art and life and the dilution of the boundaries that configure them (Dinis 2022). Through the mutual contamination between performer and spectator, some territories previously demarcated and/or at least thought of by the actor are dissolved for the construction of new, more uncertain territories. It can thus be said that performativity has brought about new configurations in the relationship between spectator and performance, leading the audience from matrices that operate beyond the narrative to aspects of physical proximity (Dinis 2022). The spectator is made aware of his participation in an artistic work, extrapolating the character of an observer to be framed as a co-participant.

In this way, performativity escapes the commonplace of everyday corporeality, creating mechanisms of continuous movement, diluting permanent boundaries, and seeking to destabilize the previously clear differences of everyday life, starting from the physical experience as a motto for spatial transgression (Fernandes 2011). In this sense, performativity acts through sounds and images, through plasticity, in the materiality of the interactions between the presentation space and the audience.

In the materiality of the interactions, the place becomes a resizing of the presentation space, endowed with sensations, affections and allusions to the lived experience, a place that retains within itself, its meaning and its dimensions of the movement of history in formation, as a movement of life, that can be grasped through memory, through the senses and the body (Carlos 1996).

Memories are important lived records that start from remembering and that eternalize places as references and scenarios for a constant visit to the past, bringing with them the most diverse feelings, documented and mentioned in narratives, imaginations and perceptions. Thus, as Nora (1993) points out, places of memory are places in the three senses of the word: material, symbolic and functional. Even a place of purely material appearance is only a place of memory if your imagination gives it a symbolic aura. They are therefore places that add a history full of complicities, meanings, affectivities and belonging.

Memory is stratified in place, searching for inscriptions and signs of absence that describe the memory of the place. As the place accumulates memories in layers that, when added together, form a unique profile, the place of memory emerges, where the community sees significant parts of its past of immeasurable affective value (Gastal 2002). The places of memory and the memories of the place, individual and collective, combine in the search for instruments to reinforce identity and singularity, thus strengthening the sense of belonging.

Memory is also inscribed over time, in the displacement between places, and the perspectives gained from immersion in these places. Places thus have a profound effect on thoughts and interpretations that arise from the way they have been felt through the body, founding the materiality of these places on aspects of representation. It is therefore a matter of giving new conceptual guidelines to the narratives of places by creating new conceptual guides.

The issue of memory and the tendency to expand its scope, considering the role of the performance of corporeal and noncorporeal practices (Hoelscher 2004), makes it so that



there is a diversity of approaches and that it is observed from several areas that look at the memory and the remembrance of the remodeled place, especially through its collective forms, to give itself a coherent identity, a narrative and a place in the world (Said 2000).

In the site-specific projects developed as part of this research, we start from the theme of memory as a phenomenon that allows the present creation of an absence (Ricoeur 2004), and we assume that any work of memory seems to imply a work of representation, which is amplified by the unique characteristics of religious places. Inherent in this work of representation is also a process of remembering that precedes a process of constructing sounds and images: sounds that are imagined to have been heard, images that are imagined to have already been visualized and sounds and images that are understood as assistants in the living experience of memory construction, promoting performativity of memory during live audiovisual performances.

It is a performativity that acts through sound and image, in the materiality of the interactions between places of memory and memories, individual and collective, place and public, in performances where the most important thing is not what the work seeks to signify or symbolize but the crossing of the experience—a crossing of the experience that goes beyond the possibility and effort of interpretation and the production of meaning, beyond pure reflection or rational interpretation, in a symbolic ritual action that mediates this performativity of memory.

During the performative moments of the site-specific projects, the performative aspects of memory are emphasized, highlighting the active and constructive nature of memory that challenges the view of memory as a passive container of past events, focusing on how memory is represented, shaped and influenced by various social, cultural and spatial factors.

In this sense, memory is interpreted as a performative action, an active and performative process rather than a simple retrieval of stored information. Memory is not seen as a static reproduction of the past but as a dynamic and creative act that involves interpretation, reconstruction and (re)contextualization. Thus, memories are shaped and influenced by the present moment in which they are reminded through an embodied perception of place. This embodied perception of place, facilitated by a performativity of the memory, can have several implications for audiences and their experiences, particularly in religious places.

Religious places often have significant cultural and historical value and embody the collective memory, ritual actions and traditions of a community. The ritual and performative actions related to the memory of and in these places reinforce a shared history and help to shape a personal and communal sense of 'self'. The embodied perception of place in religious places thus evokes deep emotional and spiritual experiences, intensifying emotional connection and facilitating a sensory experience that contributes to an immersive experience in a defined space-time. So, experiences with site-specific projects become an integral part of the site-specific projects and the meaning of these site-specific projects manifests itself in an experience (von Hantelmann 2014).

## 5. Methodology

Research through artistic practice is a process of constant questioning because, unlike other academic research models, it generates knowledge based on the experience and practice developed by artists. In this sense, since this practice is singular, unique and particular, it must be transmitted through models that correspond to its nature and through this can make use of various discursive and representational strategies.

One of the main characteristics of research through artistic practice lies in the claim that the results of research and the production of knowledge must be realized through the symbolic language produced and in the form of the practice of researcher-artists. This makes the process of research through artistic practice challenging, as the construction of any proposed approach constitutes a kind of productive uncertainty, a zone for temporary 'constructions' of concepts and contingent thinking.

The projects, developed according to a hybrid methodology and largely executed as works in progress, can be understood as an alternative form of investigative practice,

close to the recent dynamics of practice-as-research, which diverges from the context of traditional arts, driving the creation of new approaches and expanding the limits of these.

According to Witkin (2011), research is generally seen as providing important knowledge for practice, while practice can provide contextual relevance for research. However, differences in goals, language, expertise, audience and environment, among others, keep the two separate. Thus, the theme of 'practice-as-research', as adopted here, refers to beliefs and values about practice and research that create an understood gap between the current state of affairs and a more desirable panorama (Witkin 2011).

In addition to the differences mentioned by this author, three other characteristics have been presented as reasons for the separation of practice and research. These are creativity, mutability and presence. However, we defend that these characteristics can also be seen as points of convergence, as they are present in both practice and research.

To confirm this convergence, a series of site-specific research projects have been developed in religious places through artistic practice, under a format and with their language of expression, to analyze and develop functional methods and strategies linked to the development of artistic creations and the proposition of forms of their presentations. This research, through artistic practice, is developed around two main elements: the process of approaching the context of the place and the (de)construction of the sense of place, according to the approach developed by Dinis (2022).

The process of approaching the context of the place began by interacting with the places to apprehend and understand them and was carried out through permanence and several movements in them. In this sense, the site-specific projects developed in religious places are seen as a practice of memory, through which sound and visual narratives have been constructed, effective for the formulation of their corporeality and that of those who observe and receive them in the performative moments.

As a practice of memory and the materialization of this memory, each site-specific project is the result of a systematized methodology, at different times, in the approach to the place, evaluating the different levels of permanence and modalities of access to information, the relationship established with the place and the objectives defined by each site-specific project.

The methodology adopted fits into a process of research through artistic practice, since it is the practice that guides the research, and the research involves practical knowledge that can be particularly demonstrated in practice—that is, knowledge that is a matter of doing, rather than being conceived in the abstract and therefore able to be articulated only in words through a traditional research approach (Nelson 2013). This knowledge grows out of a mixture of practical and observational engagement with the beings and existences around (Ingold 2013), and its research involving works of art or artistic practice inevitably reflects an empirical dimension (Nevanlinna 2002).

This view is reinforced by Nelson (2013) when this author argues that the process of research through artistic practice involves a research project in which practice is a key method of investigation and, concerning the arts, where the practice is presented as substantive evidence of research.

Taylor (1985) suggests that these practices are semantic spaces that are indistinguishable from the language that is used to describe, invoke or perform them. These forms of research differ from the conventional methodologies traditionally recognized by the academy, precisely to be able to welcome and elaborate on questions that are intrinsically linked to the object of research and that go through several paths of formulating a hypothesis for its subsequent confirmation or refutation.

It coincides with the idea of exploring what emerges, adding that the process of research through artistic practice transcends and interweaves 'place', 'self', 'body', 'experience', 'mind', 'sensation', 'analysis', 'articulation', 'memory' and 'argument', often in idiosyncratically created structures.

This idea is also reinforced by authors such as Haseman (2006), Barrett and Bolt (2007), Kershaw and Nicholson (2011), Bonnenfant (2012), Leavy (2015) and Bala et al. (2017) when

they argue that because this artistic practice is individual, unique and particular, these models of research through artistic practice can be adapted using different approaches, as in a process of artistic research all aspects are often in motion and development (Arlander 2012). Thus, there is no general form of research that the researcher-artist can attempt to approximate, just as there is no universally accepted concept of art on which to base art-based research (Arlander 2012).

The research-creation projects of this research develop two components, the process of approaching the site-specific and the (de)construction of the sense of place, following a research approach through artistic practice (Dinis 2022), which is anchored in a 'discovery-led' research methodology (Rubidge 2005) and in concepts such as 'the undermind' (Claxton 1997, 'primary consciousness' (Edelman and Tononi 2001) and 'extended consciousness' (Damasio 2004).

The process of approaching the site-specific began with the interaction with the three religious places (Tree of Life Chapel, Chapel of the Immaculate and Church of Cedofeita) and was carried out through permanence and movements within them. The slow pace of these two actions allowed not only their registration but also the assimilation of the sensations of discovering the places, which were ordered from the memories of the places, thus highlighting the dimension of sensitive and affective experience (Jackson 1994). Observing the physical and digital records of the two actions carried out during the fieldwork, we noticed that they did not involve a subjective organization of the place but an intervention in the order of the elements presented. Thus, in this permanence and these movements, there is an intention to reorder the place and to create new local narratives, permeated with emotion, in a strategy of observing and assembling the atmosphere.

The site-specific projects focused on these three specific places, the Tree of Life Chapel, the Chapel of the Immaculate and the Church of Cedofeita, and started from the identification of elements of the context of the place, focusing on their ability to testify to symbolic aspects of the place, to inventory its memory and to reconstruct experiences of the place itself—memories and experiences that were used as guiding elements of the performative moments carried out in this research.

The process of research through artistic practice is understood by us as part of the temporal cycle of the work of art, in which its integral parts are thought together in a continuous iteration of research and action that we tend to consider appropriate to the performativity that we wish to substantiate (Dinis 2022).

The site-specific projects developed in this research were unfolded in a series of fundamental elements related to their design and realization[7]. We also consider that other information available for reading and viewing[8] is an integral and fundamental part of this essay. In addition to providing access to recordings and images of each of the creations, their consultation presents additional documentation on the process of the design, research, presentation and reception of each of the projects. Taken as a whole, these materials deepen and illustrate the paths of research through artistic practice and are therefore materials inherent to creative making itself, rightly understood as reflective practice. These three projects guarantee the thematic, temporal and spatial heterogeneity of the practical work developed within the conceptual framework of this research.

The site-specific projects were developed in two phases, the fieldwork and the creation of the sound and visual components, and followed a conceptual model of approach to place (Dinis 2022). The fieldwork covered the research process, which included reading and analyzing bibliographies about the places, interacting with these places, capturing and perceiving their environment, writing down sensations and local atmospheres, recording routines and activities in photographic and video media, sound recordings and visual recordings, and elaborating on the conceptual guide for the sound and visual component. The creation and production of the sound and visual component took place in the studio.

In the performative moments, the artist/performer is considered not only as an operator of the media that constitute, in this case, sound and image, but also as a mediator, as a creator and, consequently, as a narrator who constructs the sound and visual narratives.

At the end of each performative moment, a conversation with the audience was facilitated to get feedback on the performance and the development of the site-specific project.

From the implementation of the conceptual model of approach to place (Dinis 2022), in each of the religious places the meanings of the site-specific projects were found. These include the themes (activators of performativity) of each of the religious places chosen for the fieldwork, namely shelter (Tree of Life Chapel), humility (Chapel of the Immaculate) and fragility (Church of Cedofeita).

Inside the Conciliar Seminary of Saints Peter and Paul in Braga is the Tree of Life Chapel, a place that appeals to the senses and emotions, the result of the joint work of seminarians, teachers, architects, artists, sculptors, goldsmiths, painters, carpenters and masons. It is a wooden shelter in which the various embedded beams create a fascinating play of light and shadow, giving the chapel a luminous appearance. The public presentation of the performance[9], entitled *the slowness of waiting and echoing*, took place on 20 December 2022 in the Tree of Life Chapel of the Conciliar Seminary in Braga (Figure 1).

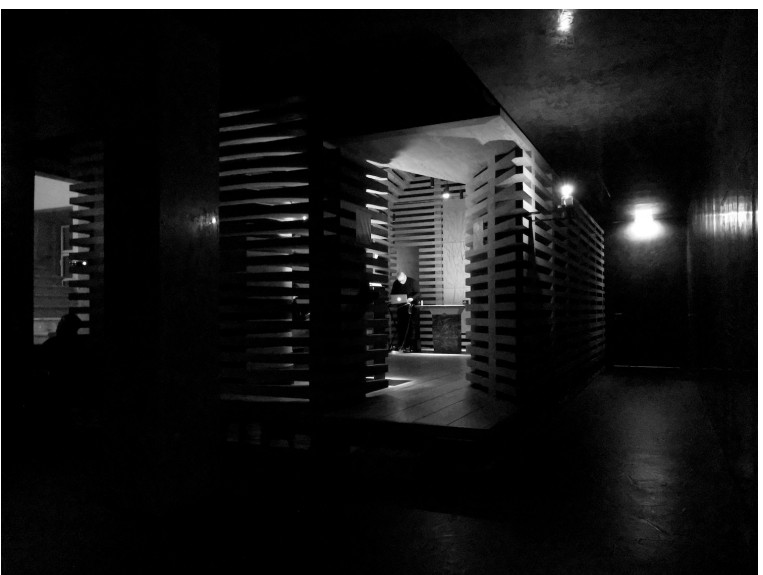

**Figure 1.** *The slowness of waiting and echoing* (20 December 2022, Tree of Life Chapel of the Conciliar Seminary, Braga).

A small forest gives access to the Chapel of the Immaculate, located in the Minor Seminary of Braga. Passing through this forest, one reaches a clearing that serves as the entrance to the assembly. All the elements of the assembly create an atmosphere of humility, conducive to introspection. There is also a 'body of light', a white marble panel suspended from a steel structure, through which an abundance of natural light floods the space, creating a qualified luminosity. The public presentation of the performance[10], entitled *the wander that aggregates and contains*, took place on 3 March 2023 in the Chapel of the Immaculate of the Seminary of Our Lady of the Conception in Braga (Figure 2).

The Church of Cedofeita is a monumental and brutal concrete structure that highlights the use of raw materials such as stone and wood. A space where time, silence and materials promote the complementarity between authenticity and fragility, and which seeks to respond to the main variables of space, time and silence, materialized in a place where autonomy and dialogue meet. The public presentation of the performance[11], entitled *the delay in searching and meeting*, took place on 2 June 2023 in the Church of Cedofeita in Porto (Figure 3).

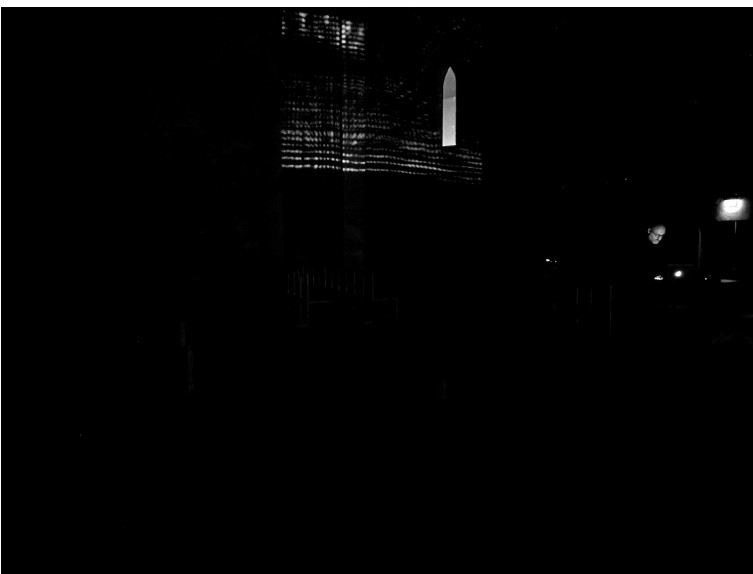

**Figure 2.** *The wander that aggregates and contains* (3 March 2023, Chapel of the Immaculate of the Seminary of Our Lady of the Conception, Braga).

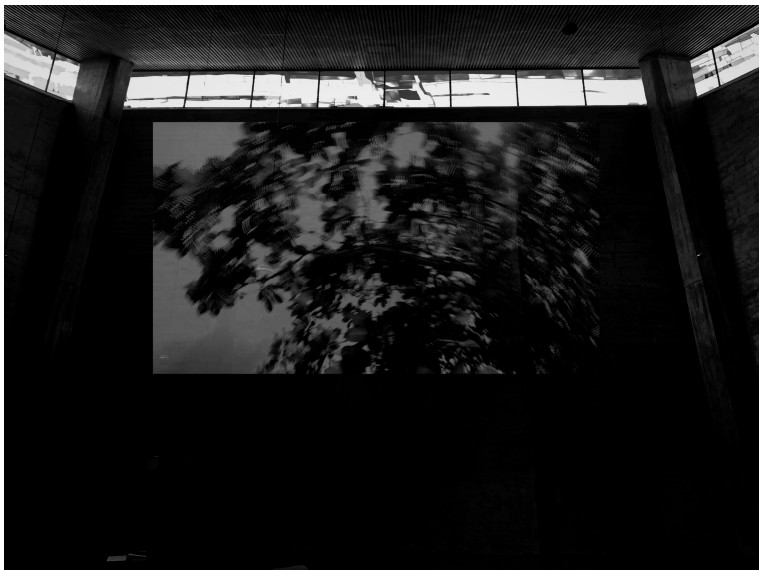

**Figure 3.** *The delay in searching and meeting* (2 June 2023, Church of Cedofeita, Porto).

Given the constant overlap between the themes of the site-specific projects (shelter, humility and fragility), the observation of the residual artifacts produced as part of the public presentations and the public feedback, we undertake that religious places are shelters for those seeking refuge, as they provide a space where one can find consolation, peace and a sense of spiritual connection, creating an atmosphere conducive to spiritual contemplation and inner reflection. These places can serve as a reminder of the importance of humility in a spiritual approach, fostering an attitude of openness, surrender and willingness to learn and grow spiritually. They are also places that inspire reflection on the fragility and impermanence of human life, inviting people to transcend their fragility and connect with something greater and more enduring, throughout the performative moments.

## 6. Conclusions

The concept of performance has been understood through different perceptions resulting from different disciplinary approaches, artistic fields or cultural contexts. Due to this conceptual openness and the diversity of creative practices, performance, as an action

before an audience, offers great potential for experimentation through the confluence of sound and visual media.

With its technological appropriation and expansion into other media, performance has stood out among the creative practices of recent decades, allowing a decentralization of the body/performer, thus opening up to other media and other materialities, such as sound or image. In the context of this decentralization, throughout this essay, we have sought to rethink and reflect on the notion of the connections between performer, spectator and religious place.

Through audiovisual mediation, we have verified that presence is produced in live performative moments and that sound and image become expressive processes in the production of these effects of presence since it is these that most obviously contribute to the production of meaning and the manifestation of the artistic work. In this way, audiovisual media should be seen as a bridge between art objects and not as an artistic destination, allowing dialogue and communication between different artistic realities.

The performative moments promote new possibilities, recursive processes, repetitions, nonlinear structures, simultaneous events and combinations of languages, where time, performative space and the performativity developed between the performer and the spectator are related with greater freedom and with the cooperation of different media, approximating them to moments that promote a symbolic liturgy.

In these live audiovisual performative moments, sound and image become convergent expressive processes, configuring the production of effects of presence and enhancing the representation of memory. The articulation between sound and live image also stimulates the creation of new narratives, making them denser and more immersive in artistic figuration, reflecting the complementarity of space and time, and opening new performative paths that are developed and experienced in live audiovisual performances, namely in religious places.

In the site-specific projects developed as part of this research, we start from the theme of memory as a phenomenon that allows the present creation of an absence, and we assume that the work of memory seems to imply a work of representation. This work of representation also involves a process of remembering that precedes a process of constructing sounds and images. These sounds and images are understood as tools in the living experience of memory construction, promoting a performativity of memory during the live sound and visual performances presented in performative moments.

During these performative moments of the site-specific projects that have been developed, the performative aspects of memory are highlighted, focusing on how memory is represented, shaped and influenced by various social, cultural, religious and spatial factors. These moments play a crucial role in the process of perception and apprehension, making use of the spatial qualities of places as they stimulate multiple senses, evoke emotional and psychological responses, take into account contextual meaning and encourage audience engagement.

The places chosen for fieldwork are spaces where particular intangible qualities can be revealed as a kind of quality that makes them special places. They are places that seek to create a spiritual connection and strengthen religious ontological positions in the world, where there is an interaction with the sacred and where the meaning and significance of human existence are intensified. This embodied perception of space in these religious places, through the performativity of memory, has an impact on the public because it promotes a sense of belonging and identity, evokes emotional and spiritual involvement, reinforces rituals and memories, allows for symbolic (re)interpretations and encourages a deeper understanding of the formation of place, our presence in religious places and our role in human life.

Throughout the development of this research and the development of site-specific projects, these (re)interpretations raised some issues about the perception and understanding of site-specific performances through the mediation of nonverbal means in religious places, to which we sought to respond.

In this sense, we observed that the performativity of memory, as an autobiographical concept, can be enhanced through live audiovisual performances in religious places by incorporating narratives and testimonies that have a connection to the place and by integrating sound and images that evoke memories and autobiographical experiences, highlighting the individual and subjective nature of memory in the spatial context of the place itself.

We have established that the performativity of memory in religious places can also promote a spatial 'self' through live audiovisual performances, creating dynamic, immersive and physical experiences in the spatial context of the religious place. This is achieved through the use of visual elements that can transform the architectural features of the place, evoke symbolism or represent narratives and iconography of the place itself, creating a deeper connection to the place and the memories related to it. The sound elements reinforce this spatial 'self', evoke emotional responses and trigger memories connected with the rituals and practices of the place. The confluence of sound and image also stimulates a multisensory experience in the here and now, reinforcing the sense of presence and connection of a spatial 'self' of the public to the place, grounding this spatiality in a contingent performative memory.

Finally, we argue that the construction of this spatial 'self' in live audiovisual performances in religious places, compelled by the performativity of memory, involves processes of social and artistic reconfiguration that contribute to transforming not only the social dynamics within the community but also the artistic representations of memory. Performative moments in religious places provide a platform for the community to shape its collective memory, contributing to the reconfiguration of social dynamics within the community and the recognition of diverse perspectives, thus fostering a sense of belonging and cohesion. These moments also involve artistic reconfigurations of practices, rituals and artistic forms, as they promote reinterpretations, representations or juxtapositions of narratives, symbolisms and religious aesthetics because of the multiple, contested, fluid and uncertain natures of these places. By reconfiguring artistic expressions, audiovisual performances stimulate creativity, artistic exploration and the formation of new aesthetic approaches that reflect contemporary sensibilities while respecting the identity of religious places.

From the residual artifacts produced and the feedback from the audience, collected during the conversations held after the performative moments, we also observed that the aesthetic and performative configurations used in audiovisual artistic creations of a place can have an impact on the most individual manifestations of religion, religiosity and religious belief, influencing the interpretation and creation of meaning, evoking emotional and spiritual responses, facilitating embedded engagement and promoting personal transformation and transcendence.

With this research through artistic practice, we also seek to anchor another sketch in the territories of performance, located between theory and practice, and the studies of religion in its multiple phenomena, traditions and grammar, questioning the aesthetic place of both today as essential tools for discovering new ways of understanding the contemporary culture and future directions on the aesthetic and performative configurations used in audiovisual artistic creations in religious places.

Due to the increasing interest in developing audiovisual artistic creations in religious places, the future directions on the aesthetic and performative configurations can include the development of immersive environments within religious places improving the experience of virtual pilgrimages or engaging in interactive narratives that enhance spiritual journeys, of interactive installations that can engage worshipers by allowing them to actively participate in the audiovisual experience. These future directions can also embrace multimedia storytelling through the use of multimedia platforms to convey stories, memories, religious narratives and personalized experiences with the use of wearable technology with curated content tailored to an individual's spiritual interests or beliefs, enhancing their engagement and understanding. Furthermore, and considering the above-mentioned conclusions, we believe it is important that future research work deepens the concept of spatial

self through its (re)construction based on ritual and belief and focusing on individual and collective remembrance processes.

**Funding:** This research received no external funding.

**Institutional Review Board Statement:** Not applicable.

**Informed Consent Statement:** Not applicable.

**Data Availability Statement:** Not applicable.

**Acknowledgments:** The Archdiocese of Braga, Conciliar Seminary of Braga, Minor Seminary of Braga, Collegiate Church and Parish of St Martin—Cedofeita, *Schola Cantorvm* Collegiate Church of Cedofeita, CITER—Research Centre for Theology and Religious Studies, Faculty of Theology of the Portuguese Catholic University, Portuguese Catholic University, CEIS20—Centre for Interdisciplinary Studies of the University of Coimbra, SELMA—Centre for the Study of Storytelling, Experientiality and Memory of the University of Turku, Institute of Education and Citizenship, Flying Thoughts—Association for the Promotion of Ideas, Braga Media Arts.

**Conflicts of Interest:** The author declares no conflict of interest.

## Notes

[1]  Available at https://fredericodinis.wordpress.com/performance/lugares-religiosos/ (accessed on 1 June 2023). This website presents documentation of the research process through artistic practice and the performative moments. Taken together, these materials deepen and illustrate the paths followed by the research presented in this essay, being for this reason inherent materials to the creative process itself, understood precisely as a reflexive practice.

[2]  These publications include *Liveness: Performance in a Mediatized Culture* (Auslander 1999) by P. Auslander; *Remediation: Understanding new media* (2000) by J.D. Bolter and R. Grusin; *Multimedia: From Wagner to Virtual Reality* (2001) by R. Packer and K. Jordon; *The New Media Book* (2002) by D. Harries; *Prefiguring Cyberculture: An Intellectual History* (2002) by D. Tofts, A. Jonson and A. Cavallaro; *The New Media Reader* (Wardrip-Fruin and Montfort 2003) by N. Wardrip-Fruin and N. Montfort; *Performance and Technology: Practices of Virtual Embodiment and Interactivity* (2006) by S. Broadhurst and J. Machon; *Intermediality in Theatre and Performance* (2006) by F. Chapple and C. Kattenbelt; *Theatre and Performance in Digital Culture: From Simulation to Embeddedness* (2006) by M. Causey; *Postdramatic Theatre* (2006) by H-T. Lehmann; *Digital Performance: A History of New Media in Theater, Dance, Performance Art and Installation* (Dixon 2007) by S. Dixon; *Closer: Performance, Technologies, Phenomenology* (2007) by S. Kozel; *Multi-Media: video–installation–performance* (2007) by N. Kaye; *A Philosophy of Computer Art* (2009) by D. Lopes; *New Media: A critical introduction* (2009) by M. Lister, J. Dovey, S. Giddings, K. Kelly and I. Grant; *Entangled: Technology and the Transformation of Performance* (2010) by C. Salter; *Mapping Intermediality* in *Performance* (2010) by S. Bay-Cheng, C. Kattenbelt, A. Lavender and R. Nelson; *Cyborg Theatre: Corporeal/Technological Intersections in Multimedia Performance* (2011) by J. Parker-Starbuck; *Performing Mixed Reality* (2011) by S. Benford and G. Giannachi; *Materializing New Media: Embodiment in information aesthetics* (2011) by A. Munster; *Multimedia Performance* (2012) by R. Klich and E. Scheer; *The Johns Hopkins Guide to Digital Media* (2014) by M-L. Ryan, L. Emerson and B. J. Robertson; *The Rhythmic Event: Art, Media, and the Sonic* (2014) by E. Ikoniadou; *Embodied Avatars: Genealogies of Black Feminist Art and Performance* (2015) by U. McMillan; *Black Performance on the Outskirts of* the *Left* (2017) by M. Gaines; *Transmission in Motion: The Technologizing of Dance* (2016) by M. Bleeker; *Performance in the Twenty-First Century: Theatres of Engagement* (2016) by A. Lavender; *The Delayed Present: Media-induced Tempor (e) alities & Techno-traumatic Irritations of "the Contemporary"* (2017) by W. Ernst; *Intermedial Theater: Performance Philosophy, Transversal Poetics, and the Future of Affect* (2017) by B. Reynolds; *Immersive Embodiment: Theatres of Mislocalized Sensation* (2019) by L. Jarvis; *Digital Theatre: The Making and Meaning of Live Mediated Performance* (2020) by N. Masura; among others.

[3]  In Beckett's *Come and Go* (1975) by the Mabou Mines, the set consisted of a huge mirror almost the full width of the stage, positioned slightly below the level of the platform and tilted upward. The actors performed on a mezzanine behind and above the audience, so that the audience could only see their own reflection in the mirror. This staging shattered the usual expectations of physical presence and contact between actors and audience (Fuchs 1985) and was characterized at the time as a 'theatre of absence', marked by the failure of the theatrical enterprise of the spontaneous word with its logocentric claims to origin, authority, authenticity, in short, of presence (Fuchs 1985).

[4]  One example is the work of The Wooster Group, in which the actors do not try to build a character but perform actions using the personal characteristics of each actor.

[5]  The creations are almost always the result of the collision, collage and assembly of multiple elements, including moving image, computer programming, light, sound and dance, resulting in a dense and highly dynamic web of overlapping text, media and performance (Dinis 2020).

6    See also Derrida (1988), Butler (1990), Bätschmann (1997), Sans (1998), Taylor (2003), Hall (1997), von Hantelmann (2010, 2014) and Féral (2015), all of whom wrote about the notion of performativity, a keyword within the discourse of contemporary art and aesthetics.

7    https://www.dropbox.com/scl/fi/4yh7wkzmj8j91cdqcuppc/p-s-doc-mapa-mental.pdf?rlkey=ocvgea2tcegrtjujav64u6keu&dl=0 (accessed on 1 June 2023).

8    Available at https://fredericodinis.wordpress.com/performance/lugares-religiosos/ (accessed on 1 June 2023).

9    The residual artifacts produced as part of the public presentation held in the Tree of Life Chapel of the Conciliar Seminary, in Braga, are available at https://fredericodinis.wordpress.com/2022/12/21/capela-arvore-da-vida/ (accessed on 1 June 2023).

10   The residual artifacts produced as part of the public presentation, held in the Chapel of the Immaculate of the Seminary of Our Lady of the Conception in Braga, are available at https://fredericodinis.wordpress.com/2023/03/06/capela-da-imaculada/ (accessed on 15 June 2023).

11   The residual artifacts produced as part of the public presentation, held at the Church of Cedofeita, in Porto, are available at https://fredericodinis.wordpress.com/2023/06/03/igreja-de-cedofeita/ (accessed on 16 June 2023).

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
