# Peer review of "Performativity of the Memory of Religious Places through Sound and Image"

_religions, doi:10.3390/rel14091137_

Round 1

Reviewer 1 Report

It's not certain what the argument of this paper is – whether its possible that religious sites can be a place of reconstruction or to propose that practice-as-research could be a strong methodology to investigate this. However, no clear methodology is proposed in the end and/or no examples of the novelty of the practice are including.

Using a case study or articulating the author's own practice-as-research and the results and findings of that would make a much stronger paper. It makes reference to examples in the paper, but there is no sense of what the author(s) actually produced – only a general overview with some wider notes on where the performances happened.

Other points that could be strengthened include:

'Performativity' is not defined in an academic concept but assumed to mean 'performance-like'. However the academic history of this term is about 'doing' (e.g. Austin 'How to Do Things With Words' but more recently Von Hantelamann - How to Do Things with Art (2010) and a PhD thesis from Taylor 2017 – available here – http://repository.falmouth.ac.uk/3207/). Performativity is a doing, which could be framed as this reconstruction of spatial self but needs to be articulated in those terms

People like Ranciere are used but there is more relevant and recent literature on spectatorship and performance

How much of the spatial self is reconstructed through ritual and faith? This could be informed by repetition and nostalgia

Performing ritual and its presence/absence is interesting here 

And presence/absence may also be informed by Derrida

You could also include documentation of your work and its findings such as images - if it is about practice-and-research then it needs to be more psecifically about your project and what is new/novel about your particular methodology in invoking the spatial self in religious places.

Reviewer 2 Report

This paper presents a very interesting approach on memory, in its articulation with audiovisual performance. Exploring the place(s) of worship and/or religious practice itself, and the role of audiovisual performance as a possible spiritual/religious mediator, could enrich the project. 

About the main question addressed by the research, I believe it's quite clear that the focus is the exploration of the confluence between site-specific audiovisual performances and religious places.

Nowadays we assist to various emerging artistic practices and the site-specific is one of those. As an specific aesthetic, site-specific sound and visual performances are very connected with the place, and its characteristics. The place, in its most varied and possible interpretations, became a part of the artwork/audiovisual performance, making each artwork unique and exclusive. Site-specific is common in places with strong symbolism such as historic sites and monuments. This work reveals some originality due to the fact that it focuses on places of worship, such as churches and chapels.

About the conclusions:  

"impact on the public promotes a sense of belonging and identity, evokes emotional and spiritual involvement, reinforces rituals and memories..."

"we also observed that the aesthetic and performative configurations used in audio visual artistic creations of a place can have an impact on the most individual manifestations of religion, religiosity and religious belief, influencing the interpretation and creation of meaning, evoking emotional and spiritual responses, facilitating embedded engagement and promoting personal transformation and transcendence"

The author seems to assume the audiovisual performance as ritual itself? Is the artist and/or the performance a kind of liturgical inducer or an integral part of the religious ritual?

Also, the author should better specify how he drew the conclusions he refers to about the perceptions/experiences of the audience and also mention what distinguishes site-specific artworks in religious places from other equally symbolic places.

In general, the paper is well written, although some sentences/expressions apparently translated from other language can still be improved.

English is acceptable, although some sentences/expressions directly translated from the native language can still be improved. 
